# In-Silico Investigation of Effects of Single-Nucleotide Polymorphisms in PCOS-Associated CYP11A1 Gene on Mutated Proteins

**DOI:** 10.3390/genes13071231

**Published:** 2022-07-12

**Authors:** Fatima Muccee, Osama Bijou, Steve Harakeh, Rabi’atul Adawiyah, R. Z. Sayyed, Leila Haghshenas, Dikhnah Alshehri, Mohammad Javed Ansari, Shakira Ghazanfar

**Affiliations:** 1School of Biochemistry and Biotechnology, University of Punjab, Lahore 52254, Pakistan; 2Obstetrics and Gynaecology Department, Faculty of Medicine (FM), King Abdulaziz University, Jeddah 21589, Saudi Arabia; dr_bajouh@yahoo.com; 3King Fahd Medical Research Center, King Abdulaziz University, Jeddah 21589, Saudi Arabia; sharakeh@gmail.com; 4Yousef Abdul Latif Jameel Scientific Chair of Prophetic Medicine Application, Faculty of Medicine (FM), King Abdulaziz University, Jeddah 21589, Saudi Arabia; 5Faculty of Health and Life Sciences, INTI International University, Nilai 71800, Negeri Sembilan, Malaysia; rabiatul.ahmad@newinti.edu.my; 6Department of Microbiology, P.S.G.V.P. Mandal’s S I Patil Arts, G B Patel Science and S.T.K.V.S. Sangh Commerce College, Shahada 425409, India; sayyedrz@gmail.com; 7Department of Molecular Genetics, Postdoc Association Member of Harvard Medical School, Boston, MA 02138, USA; leilahagh@yahoo.com; 8Department of Biology, Faculty of Science, Tabuk University, Tabuk 71491, Saudi Arabia; dalshehri@ut.edu.sa; 9Department of Botany, Hindu College Moradabad, Mahatma Jyotiba Phule Rohilkhand University, Bareilly 244001, India; mjavedansari@gmail.com; 10National Institute for Genomics Advanced Biotechnology (NIGAB), National Agricultural Research Centre (NARC), Islamabad 45500, Pakistan; shakira_akmal@parc.gov.pk

**Keywords:** steroidogenesis, hirsutism, multifactorial, hyperandrogenism, single nucleotide polymorphism, translation

## Abstract

Polycystic ovary syndrome (PCOS) is a reproductive disorder with multiple etiologies, mainly characterized by the excess production of androgens. It is equally contributed to by genes and environment. The CYP11A1 gene is imperative for steroidogenesis, so any dysregulation or mutation in this gene can lead to PCOS pathogenesis. Therefore, nucleotide diversity in this gene can be helpful in spotting the likelihood of developing PCOS. The present study was initiated to investigate the effect of single nucleotide polymorphisms in human CYP11A1 gene on different attributes of encoded mutated proteins, i.e., sub-cellular localization, ontology, half-life, isoelectric point, instability index, aliphatic index, extinction coefficient, 3-D and 2-D structures, and transmembrane topology. For this purpose, initially coding sequence (CDS) and single nucleotide polymorphisms (SNPs) were retrieved for the desired gene from Ensembl followed by translation of CDS using EXPASY tool. The protein sequence obtained was subjected to different tools including CELLO2GO, ProtParam, PHYRE2, I-Mutant, SIFT, and PolyPhen. It was found that out of seventy-eight SNPs analyzed in this project, seventeen mutations, i.e., rs750026801 in exon 1, rs776056840, rs779154292 and rs1217014229 in exon 2, rs549043326 in exon 3, rs755186597 in exon 4, rs1224774813, rs757299093 and rs1555425667 in exon 5, rs1454328072 in exon 7, rs762412759 and rs755975808 in exon 8, and rs754610565, rs779413653, rs765916701, rs1368450780, and rs747901197 in exon 9 considerably altered the structure, sub-cellular localization, and physicochemical characteristics of mutated proteins. Among the fifty-nine missense SNPs documented in present study, fifty-five and fifty-three were found to be deleterious according to SIFT and PolyPhen tools, respectively. Forty-nine missense mutations were analyzed to have a decreasing effect on the stability of mutant proteins. Hence, these genetic variants can serve as potential biomarkers in human females for determining the probability of being predisposed to PCOS.

## 1. Introduction

Polycystic ovary syndrome (PCOS) is a disease of the human female reproductive system associated with physiological, psychological, metabolic, endocrine, and menstrual malaises. The disorder is known as polycystic due to detection of many undeveloped follicles in ovary called as cysts in ultrasonography (USG). Cysts arise from primitive follicles which failed to develop due to abnormal ovarian function caused by hormonal disturbances [1]. PCOS affects 6–12% of females at their reproductive age, i.e., 15–49 years [2]. According to another study, the amount of females affected with PCOS is 4–20% [3]. Recently, a study was conducted on a population of 1960 eligible Iranian females to detect PCOS prevalence using three diagnostic criteria, i.e., National Institutes of Health (NIH), the Androgen Excess Society (AES), and the 2003 Rotterdam [4,5]. According to these criteria, the prevalence rates have been determined to be 13.6%, 17.8%, and 19.4%, respectively [6]. Another study performed to detect the global incidence of PCOS among females of susceptible age revealed 1.55 million PCOS cases with uncertainty intervals (UIs) of 95% [7].

Reproductive abnormalities are fluid-filled sacs in enlarged ovaries (polycystic ovaries), endometrial carcinoma, complete absence of menstruation at puberty or amenorrhea, uterine and abdominal cramps during the reproductive cycle also known as dysmenorrhea, and the absence of the release of ovum from the ovary due to a decreased amount of luteinizing hormone (LH) and follicle-stimulating hormone (FSH) associated with estrogen deficiency or anovulation [1]. Physiological symptoms include obesity, acne, thinning of hair, alopecia, hirsutism, and heavy hair growth on the face [8]. Psychological symptoms are characterized by decreased self-esteem, sexual satisfaction, depression, anxiety, and a negative perception of body image [9,10]. In aged females, the disorder undergoes a transition from reproductive to a more metabolic one. The metabolic symptoms are dyslipidemia, hypertension, cardiovascular diseases, hepatic steatosis, type-2 diabetes (T2DM), impaired glucose tolerance (IGT), and increased resistance to insulin [10,11].

Under normal conditions, a sex hormone-binding globulin protein (SHBG) binds testosterone and controls its proportion in body circulation. However, high concentration of insulin inhibits SHBG, leading to an excess amount of free circulating male sex hormone androgens which, in turn, increases hypothalamic gonadotropin-releasing hormone (GnRH) [12,13]. It leads to more secretion of gonadotropin hormone (GH) from the pituitary gland. This causes a rise in luteinizing hormone (LH) concentration over follicle-stimulating hormone (FSH). LH binds with the LH receptor and promotes ovarian androgenic production [14,15]. Over-production of anti-mullerian hormone (AMH) from granulosa cells of human ovary causes inhibition of the development of primitive follicles, resulting in the formation of cysts [16].

PCOS is a multifactorial trait being regulated by interactions of polygenes and the environmental factors. The environmental factors that may contribute to PCOS may be physical, chemical, or related to diet and lifestyle, etc. Chemical factors are exposure to mercury, cadmium, lead, pesticides, carbon disulfide, organic solvents such as toluene, ether and ethylene glycol, di-isobutyl phthalate (DIBP), di-isononyl phthalate (DINP), and areca nut chewing and tobacco smoke [17,18,19]. Other external factors include stress and mood swings, difficulty in weight loss, weight gain, sleep apnea, intake of high-calorie food, and the use of plastic ware and indoor decorations [20,21].

A wide range of genes with functional diversity are reported to be associated with PCOS, including gene-encoding sex hormone-binding globulin protein (*SHBG*), gene-encoding androgen receptor (*AR*), gene-regulating adrenal and ovarian steroidogenesis, i.e., *cytochrome P450, family 11, subfamily A, member 1* (CYP11A1), *cytochrome P450, family 21, subfamily A, member 2* (CYP21A2), *cytochrome P450, family 17, subfamily A, member 1* (CYP17A1), *cytochrome P450, family 1, subfamily A, member 1* (CYP1A1), *cytochrome P450, family 21, subfamily A, member 2* (CYP21A2), *cytochrome P450, family 3, subfamily A, member 7* (CYP3A7), and *cytochrome P450, family 19, subfamily A, member 1* (CYP19A1), genes associated with regulation and function of gonadotropin hormones such as *follistatin* (FST) gene and gene-encoding β-subunit of luteinizing hormone (LHβ) [22,23,24,25,26,27,28,29,30,31,32], and genes regulating the function and production of insulin, i.e., *insulin* (INS) gene, *insulin receptor* (INSR) gene, *insulin-like growth factor 1* (IGF), *calpain-10 gene* (CAPN10), and *insulin receptor substrate genes* (IRSs) [33,34,35,36]. In addition to this are chronic inflammation-associated genes including *interleukin-6* (IL6), *interleukin-6 receptor α* (IL-6 receptor-α) and *IL-6 signal transducer* (IL-6ST), *tumor necrosis factor α* (TNF-α), and *tumor necrosis factor 2* (TNFR2) [37,38,39,40], genes involved in the signaling of availability of metabolic fuels and energy homeostasis, i.e., *leptin* (ob) gene, *leptin receptor gene* (LEPR), *adiponectin gene* (ADIPOQ), *peroxisome proliferator activated receptor γ* (PPAR-γ), *and peroxisome proliferator-activated receptor γ coactivator* (PGC-1α) [41,42,43,44]. In addition to these, some other genes have also been found to be the cause of PCOS, i.e., *plasminogen activator inhibitor-1* gene (PAI-1), *apoprotein E* (ApoE), *resistin* (RETN), *glycogen synthetase* (GYS1), *paraoxonase* (PON), *aldosterone synthetase* (CYP11B2), *dopamine receptor* (DRD), *17α-hydroxysteroid dehydrogenase* (HSD), and *3-β-hydroxysteroid dehydrogenase* (HSD3B2) genes [33,38,45,46,47,48,49,50,51,52].

CYP11A1 belongs to the cytochrome P450 family. It is alternatively known as cholesterol side chain cleavage enzyme, cholesterol desmolase, CYPXIA1, cytochrome P450 11A1, and cytochrome P450 (scc). This gene is functionally related with synthesis of steroid hormones. It encodes enzyme which catalyzes the transformation of cholesterol into pregnenolone (a precursor of steroid hormones) in mitochondria. This is the very first step of the steroidogenesis pathway [52].

High concentrations of CYP11A1 or any mutational change in protein may enhance the steroidogenesis, resulting in hyperandrogenism which contributes to PCOS pathogenesis. The literature provides detailed data regarding CYP11A1 gene polymorphisms strongly associated with PCOS, including SNP rs4077582, a pentanucleotide repeat polymorphism (tttta)n in the promoter region, rs743572, rs4887139, and rs4886595 [53,54,55,56,57].

SNPs in CYP11A1 can be useful in establishing the genetic architecture of PCOS. However, the literature reports so far are scarce. Therefore, keeping in view the significant association of CYP11A1 gene with PCOS, the present project was designed in order to analyze the effect of the reported SNP’s transcript variant of this gene on the properties of the encoded enzyme. These SNPs might be useful in prediction and susceptibility of PCOS in human females and can be tested further through experimental work.

## 2. Materials and Methods

In present study, in-silico characterization of the CYP11A1 gene was performed to predict the effect of the reported SNPs on the encoded mutated proteins. The scheme is shown in Figure 1.

### 2.1. Retrieving CDS of CYP11A1

CYP11A1 comprises 10 transcripts. A transcript variant of this gene, i.e., CYP11A1201 (ENST00000268053.11) located on chromosome GRCh38.p12 was focused on in this study. This splice variant is a product of gene ENSG00000140459.18. This transcript comprises 9 exons encoding a protein of 521 amino acids. Coding sequence and SNPs of this variant were retrieved from the ENSEMBL database (https://asia.ensembl.org/index.html, accessed on 23 March 2022).

### 2.2. Retrieving SNPs and Construction of Mutated CDS

The ENSEMBL database was used for retrieving the SNPs. A total of seventy-eight SNPs were retrieved in the exonic regions (Table 1).

A total of twenty-two cases were designed for seventy-eight SNPs. SNPs were analyzed through the SIFT tool (https://sift.bii.a-star.edu.sg, accessed on 25 June 2022) and PolyPhen (genetics.bwh.harvard.edu/pph2/, accessed on 25 June 2022) to predict the nature of the effect that may be deleterious or benign.

For each case documented in present study, separate mutated CDSs were prepared by incorporating mutations in CDSs of wild type gene sequence. The location of the mutations discussed in the present study are shown in detail in Figure 2.

### 2.3. Translation of CDS into Protein Sequence

The EXPASY translation tool (https://web.expasy.org/translate, accessed on 23 March 2022) was used to translate the wild type and mutated CDSs. A total of twenty-two mutated protein sequences were obtained for all the cases designed (Appendix A).

### 2.4. Effect of SNPs on Stability of Mutated Proteins

I Mutant Suite (https://gpcr.biocomp.unibo.it/cgi/predictors/I-Mutant2.0/I-Mutant2.0.cgi, accessed on 25 June 2022) was used to predict the effect of all missense variations on the stability of mutated proteins.

### 2.5. Sub-Cellular Localization and Ontology of Wild Type and Mutated Proteins

To study the effect of SNPs in each case on the localization and ontology of mutated proteins, the CELLO2GO online tool was used [58]. The results obtained were compared with the localization and ontology of normal protein.

### 2.6. Physicochemical Features

ProtParam tool (https://web.expasy.org/protparam/, accessed on 25 March 2022) was used to analyze the effect of SNPs in question on the physicochemical parameters of mutated proteins. The parameters include number of amino acids, molecular weight, theoretical pI, half-life, instability index, aliphatic index, and extinction coefficient. 

### 2.7. 3-D and 2-D Structures and Trans-Membrane Topology

The PHYRE2 tool (http://www.sbg.bio.ic.ac.uk/~phyre2/html/page.cgi?id=index, accessed on 15 April 2022) was used to determine the effect of SNPs on 2-D and 3-D structures of mutated proteins and trans-membrane topology.

## 3. Results

### 3.1. Effect of SNPs on Mutated Protein Stability

Fifty-nine missense SNPs addressed in the present study were examined for their influence on the stability of proteins. It was found that all the missense mutations except rs1258660118, rs1393077247, rs1424340465, rs139449608, rs867506250, rs1208632679, rs752776256, rs1416463682, and rs200726137 contributed to the destabilization of the corresponding mutated proteins (Table 2).

### 3.2. Effect of SNPs on Sub-Cellular Localization of Proteins

Mutations documented in cases 2, 9, 10, 11, 14, 15, 16 to 19, 21, and 22 changed the localization of the corresponding mutated proteins from mitochondrial to higher in cytoplasm and high in mitochondria. In case 20, mutation significantly affected the location of the mutant protein and changed it to cytoplasmic only. Mutations included in cases 3, 5, 12, and 13 changed localization from mitochondrial to higher in mitochondria and high in cytoplasm. Mutations documented in cases 6, 7, 8, and 9 did not affect the localization. Case 4 mutation changed localization from mitochondrial to highest in the nucleus, higher in cytoplasm, and high in mitochondria. Case 1 changed the localization to higher in mitochondria and high in the nucleus (Figure 3, Table 3).

### 3.3. Effect of SNPs on Ontology of Mutated Proteins

Ontology analysis revealed the effect of SNPs on three types of molecular and biological functions. The molecular functions included ion binding, oxidoreductase activity, and lipid binding while biological functions discussed are lipid metabolism, biosynthesis, and small-molecule metabolism. No prominent effect was observed on the ion binding and oxidoreductase activity in cases 16 to 22. Slight or no deviations from normal values were observed with reference to lipid binding, lipid metabolism, biosynthesis, and small-molecule metabolism in cases 17 to 22 (Appendix A). Largest deviation from normal values of ion binding and oxidoreductase activity was recorded in cases 1, 9, 14, and 15. Case 1 showed the highest deviation from lipid binding followed by case 4, 3, 7, 8, 10–13, 15, and 16. As far as lipid metabolism, biosynthesis, and small molecule metabolism are concerned, the greatest variation was observed in cases 1, 3, and 4 (Figure 4, Table 4).

### 3.4. Effect of SNPs on Physicochemical Parameters of Mutated Proteins

A smaller number of amino acids and lower molecular weight in mutated proteins as compared with normal values was observed due to mutations introduced in case 3 (rs776056840 in exon 2), case 6 (rs549043326 in exon 3), case 1 (rs750026801 in exon 1), case 5 (rs1217014229 in exon 2), case 4 (rs779154292 in exon 2), case 7 (rs1421587886 in exon 3), case 8 (rs1178589612 in exon 8), case 10 (rs755186597 in exon 4), case 11 (rs1224774813 in exon 4), case 12 (rs757299093 in exon 5), case 13 (rs1555425667 in exon 5), case 15 (rs1454328072 in exon 7), case 16 (rs762412759 in exon 8), and case 17 (rs75597808 in exon 17) (Table 5).

All the physicochemical characteristics were affected by mutations incorporated in different exons of the gene. The highest variation in isoelectric point (pI) was observed in cases 1, 6, 8, and 4 due to SNPs rs750026801 in exon 2 (case 1), rs549043326 in exon 3 (case 6), rs1178589612 in exon 3 (case 8), and rs779154292 in exon 2 (case 4). Deviation noted in these cases was 10.52, 9.59, 9.56, and 9.33, respectively. The half-life of mutated proteins remained unaffected in all cases, i.e., 30 hours.

The highest variation in instability index, i.e., 52.01, 51.45, 50.14, 43.58, and 43.24 was observed in cases 4, 1, 5, 7, and 8, respectively. This variation was contributed to by SNP rs776056840 documented in exon 2 (case 4), single-frame shift mutation rs750026801 introduced in exon 1 (case 1), single-stop gained mutation rs1217014229 in exon 3 (case 5), single-frame shift mutation rs1421587886 in exon 3 (case 7), and single-frame shift mutation rs1178589612 introduced in exon 3 (case 8).

The aliphatic index of mutated proteins was affected by single nucleotide variations in case 1 and case 4 due to the SNPs rs750026801 and rs776056840 incorporated in exon 2. The values observed were 54.86 and 70.89. Other SNPs noticed in the remaining cases also affected the aliphatic index but the variation was not considerable.

As far as the extinction coefficient is concerned, SNPs analyzed in cases 2, 14, and 18 to 22 were not affected much. However, SNPs documented in cases 3 (rs776056840 in exon 2), 4 (rs779154292 in exon 2), 1 (rs750026801 in exon 1), 5 (rs1217014229 in exon 2), 6 (rs549043326 in exon 3), 10 (rs755186597 in exon 4), 7 (rs1421587886 in exon 3), 8 (rs1178589612 in exon 3), 11 (rs1224774813 in exon 4), and 12 (rs757299093 in exon 5) caused a change in the extinction coefficients of mutated proteins (Table 4).

### 3.5. Effect of SNPs on 3D Structures of Mutated Proteins

In case 1, a single-frame shift mutation rs750026801 in exon 1 was documented, which caused a change in amino acid at the 85 position of the mutated protein as well as the reading frame of the sequence. This mutation caused abnormality in the overall 3D conformation of protein. Fifteen missense mutations incorporated in exons 1, 2, and 3 in case 2, nineteen missense mutations documented in exons 4, 5, and 6 in case 9, and twenty four missense SNPs incorporated in exons 7, 8, and 9 in case 14 did not affect the overall 3D conformation of mutated protein (Appendix A).

In case 3, single-stop gained mutation rs776056840 in exon 2 was introduced in exon 2. It caused a change of the encoding codon to stop the codon at position 120 of the protein sequence, resulting in a truncated protein. In case 4, single-frame shift mutation rs779154292 was incorporated in exon 2 of CDS, resulting in a change of amino acid I > X at position 102. This led to the abnormal protein conformation. In case 5, single-frame shift mutation rs1217014229 was documented in exon 2 which caused a change in amino acid H > PX at position 130 of the protein sequence. This change resulted in the distortion of the overall protein structure. In case 6, single-stop gained mutation rs549043326 was introduced in exon 3, which caused a change at codon 144, converting it into a stop codon. This resulted in the truncation of the protein. In case 7, single-frame shift mutation rs1421587886 was incorporated in exon 3, which resulted in change of amino acid A > X at position 173 of the protein sequence. As a consequence, the 3D protein structure was altered. In case 8, single-frame shift mutation rs1178589612 was incorporated in exon 3 of the gene. This mutation replaced the amino acid L > X at the 170 position of the protein sequence, which led to an alteration in the overall protein structure. In case 10, single-stop gained mutation rs755186597 was documented in exon 4 of CDS, which caused a change in the encoding codon into a stop codon at position 232 of the protein sequence. This led to the formation of a truncated protein.

In case 11, single-stop gained mutation incorporated in exon 5 of gene resulted in a stop codon at position 282 of the protein sequence, resulting in the formation of a truncated protein. In case 12, a frame-shift mutation rs757299093 documented in exon 5 of CDS changed the reading frame of mRNA, leading to the formation of an abnormal protein. In case 13, a mutation rs1555425667 documented in exon 5 changed the amino acid E > X at position 314 of the protein sequence as well as the reading frame, resulting in the formation of a distorted protein structure. In case 15, a stop-gained mutation rs1454328072 in exon 7 of gene resulted in a change of the coding sequence at 405 to a non-coding sequence. This led to the formation of a truncated protein.

Stop gained mutations rs762412759 (case 16) and rs75597808 (case 17) documented in exon 8 of gene resulted in stop codon formation at 424 and 439 in the amino acid sequence, respectively. This led to the truncation of the mutated protein. Five stop-gained mutations incorporated in exon 9 of the gene, i.e., rs754610565 (case 18), rs779413653 (case 19), rs765916701 (case 20), rs1368450780 (case 21), and rs747901197 (case 22) resulted in the formation of stop codons at positions 521, 522, 488, 516, and 517, respectively. As a result, truncated proteins were formed (Figure 5).

### 3.6. Effect of SNPs on Secondary Structure of Mutated Proteins

Three characteristics of secondary structures, i.e., disorder, α helix, and β strand percentage were analyzed. SNPs documented in case 1 (rs750026801 in exon 1), case 3 (rs776056840 in exon 2), case 4 (rs779154292 in exon 2), case 5 (rs1217014229 in exon 2), and case 6 (rs549043326 in exon 3) caused larger deviations from normal disorder values in mutated proteins. On the other hand, mutations incorporated in case 2 and cases 7 to 22 did not affect the disorder (%) of mutated proteins.

Mutations documented in cases 1, 6, 7, 8, 11, 12, 13, and 15 markedly changed the α helix (%) in mutated proteins, i.e., 38%, 43%, 62%, 63%, 62%, 63%, and 62%, respectively, as compared with the normal value, which is 50%. SNPs in cases 7, 8, 10, 11, 12, 13, and 15 changed their β strand percentage markedly in mutated proteins. Mutations addressed in cases 2, 9, 14, 18, 19, 21, and 22 did not change β strand composition at all (Appendix A).

### 3.7. Effect of SNPs on Trans-Membrane Topology of Mutated Proteins

In case 1, case 3, case 4, case 5, case 6, and case 8, no topology was observed through in-silico analysis. Mutation induced in case 3 resulted in the formation of three trans-membrane domains in mutant form as compared with two domains in normal protein. In cases 2, 9, and 11 to 22, mutations did not affect the number of domains of mutated proteins (Appendix A). Mutations addressed in cases 7 and 10 reduced the number of domains to 1 in mutant forms of protein compared with normal (Figure 6). A slight variation in the number of amino acids in signal peptide and trans-membrane segments of mutated proteins was also observed (Appendix A).

### 3.8. Nature of SNPs Effect

Two tools, i.e., SIFT and PolyPhen helped us to determine the nature of SNPs (Table 1). According to SIFT, mutations rs750026801, rs533078157, rs143655263, rs1316467116, rs1393077247, rs1440123283, rs1207802955, rs1416104210, rs1596159786, rs1228084259, rs746124429, rs1459305798, rs1046646548, rs1567051424, rs867506250, rs1220602604, rs200029503, rs1321165216, rs566280511, rs754698583, rs551306530, rs777925426, and rs200726137 were tolerated. However, the SNPs rs1444841908, rs1402720332, rs746842413, rs1471344818, rs546976108, rs190764523, rs866217710, rs1258660118, rs762299364, rs1400474918, rs1567052760, rs1461423064, rs1021942880, rs1596159365, rs747101738, rs1402190131, rs748120824, rs1424340465, rs139449608, rs531531464, rs745719036, rs867506250, rs1208632679, rs752776256, rs1398357296, rs944327325, rs771663597, rs1448075161, rs1344040376, rs1416463682, rs780138488, rs774799229, rs757055824, rs758061011, rs1326008763, rs775664050, rs750026801, rs776056840, rs779154292, rs1217014229, rs549043326, rs1421587886, rs1178589612, rs755186597, rs1224774813, rs757299093, rs1555425667, rs762412759, rs755975808, rs754610565, rs779413653, rs765916701, rs1368450780, rs747901197, and rs148124218 were found to have deleterious effects.

According to PolyPhen, mutations rs750026801, rs533078157, rs143655263, rs1316467116, rs1258660118, rs1393077247, rs1440123283, rs1207802955, rs1416104210, rs1596159786, rs1459305798, rs747101738, rs745719036, rs1208632679, rs1220602604, rs200029503, rs1321165216, rs1448075161, rs566280511, rs754698583, rs551306530, rs777925426, and rs200726137 were categorized as benign. However, the SNPs rs1444841908, rs1402720332, rs746842413, rs1471344818, rs546976108, rs190764523, rs866217710, rs762299364, rs1400474918, rs1567052760, rs1461423064, rs1228084259, rs1021942880, rs746124429, rs1596159365, rs1402190131, rs748120824, rs1424340465, rs139449608, rs1046646548, rs1567051424, rs531531464, rs752776256, rs1398357296, rs944327325, rs771663597, rs1344040376, rs1416463682, rs780138488, rs774799229, rs757055824, rs758061011, rs1326008763, rs775664050, rs750026801, rs776056840, rs779154292, rs1217014229, rs549043326, rs1421587886, rs1178589612, rs755186597, rs1224774813, rs757299093, rs1555425667, rs762412759, rs755975808, rs754610565, rs779413653, rs765916701, rs1368450780, rs747901197, and rs148124218 were possibly damaging.

## 4. Discussion

According to the literature, increased expression of the CYP11A1 gene causes enhanced steroidogenesis in ovaries of PCOS patients. Therefore, it is considered a potential candidate gene for PCOS [59]. Association studies of this gene have been performed in PCOS-infected British, Greek, Indian, Chinese, Iraqi, and Spanish females. In Iraqi females, PCOS was found to be associated with the incidence of three and five repeats in the promoter region of CYP11A1 [60].

The PCOS-infected Caucasian females of America have been reported to exhibit pentanucleotides with nine repeats. On the other hand, Chinese and Spanish females with eight and four repeats, respectively, have been reported [61,62].

A study conducted in the Indian population revealed the connection between PCOS and a pentanucleotide polymorphic repeat. Females with more than eight repeats were infected with PCOS while those with less than eight were found healthy [63].

Most of the studies conducted so far dealt with SNPs located in the promoter region and UTRs of the gene. A study conducted on Chinese patients of PCOS reported the association of seven SNPs, i.e., rs1843090, rs11632698, rs4887139, rs4077582, D15S1547, D15S1546, and D16S520, with this disease [64]. A strong association of rs4077582 was also reported in another study. This SNP was found to induce a change in the level of testosterone by affecting LH production [53,65].

A strong correlation between severe hyperandrogenaemia and (TTTA)n microsatellite (-528 pairs) was reported in Greek patients [66].

Another study reported a strong relation of pentanucleotide repeat (TTTTA) found in the 5′UTR region of the gene with abnormally high testosterone level in PCOS-infected British females [67].

Another investigation targeted this microsatellite pentanucleotide polymorphism and found it to be the cause of enhanced risk of PCOS in Egyptian females [67,68,69]. A project aimed at finding the association of CYP11A1 polymorphisms with PCOS revealed three potentially effecting genetic variations, i.e., rs4887139, rs4077582, and rs11632698 [70].

Hence, the literature strongly supports association of CYP11A1 gene polymorphism with incidence of PCOS.

Although mutational changes in promoter and untranslated regions might affect the transcription regulation of the gene [71], mutations in exons are more crucial because they may directly alter the structure and function of the protein. Therefore, the present study was initiated to determine the effect of polymorphisms localized in exons of this gene. For this purpose, polymorphisms were obtained from a genome-wide database of human variations produced by the HapMap project and investigated for their effect on protein structure and activity.

Mutations rs776056840 (exon 2), rs549043326 (exon 3), rs755186597 (exon 4), rs1224774813 (exon 5), rs1454328072 (exon 7), rs762412759 and rs755975808 (exon 8), rs754610565, rs779413653, rs765916701, rs1368450780, and rs747901197 (exon 9) caused truncation of protein. Other polymorphisms that altered the structure of protein included rs750026801 (exon 1), rs779154292 and rs1217014229 (exon 2), and rs757299093 and rs1555425667 (exon 5). These mutations were found to have a strong impact on the nature of encoded proteins and, hence, can be suggested to be considered for PCOS association studies as well as strong genetic biomarkers for the disease. These SNPs can also be helpful in determining the susceptibility of females for PCOS.

## 5. Conclusions

By comparing the literature as well as the present study regarding the CYP11A1 gene, it can be inferred that both the promoter and the exonic regions are prone to PCOS-related mutations. Exonic mutational changes may lead to the formation of proteins with faulty structure and function. In addition, polymorphism in the promoter region may contribute to up- or downregulation of gene expression. Both these cases may have deleterious consequences. Therefore, study of the SNPs in candidate genes of PCOS such as CYP11A1 might be helpful in the future as genomic markers for PCOS susceptibility, in tracking the inheritance pattern of these variants in families, and for the elucidation of the role of genes in disease pathogenesis.

## Figures and Tables

**Figure 1 genes-13-01231-f001:**
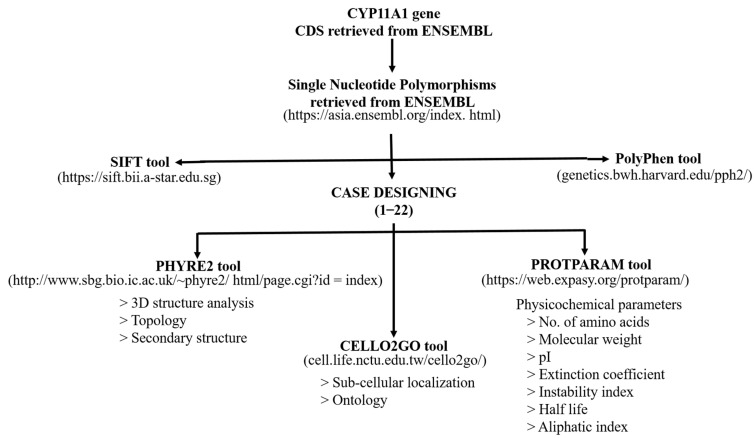
A schematic representation of the in-silico analysis performed in the present study.

**Figure 2 genes-13-01231-f002:**
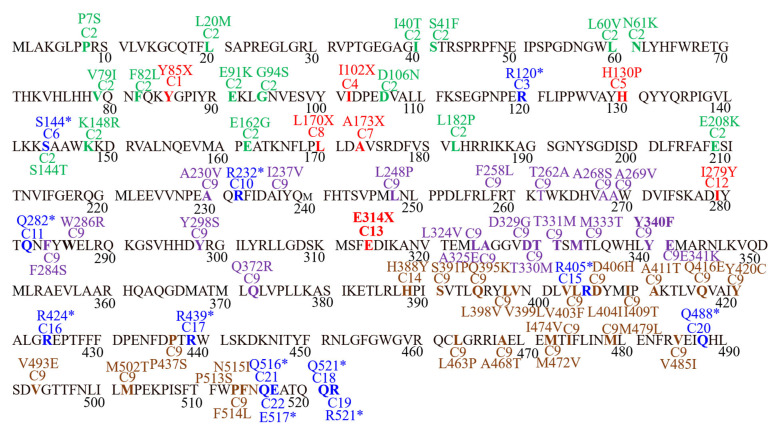
Graphical presentation of SNPs, their position, and effect documented the in present study. Red color = cases documenting frameshift mutations, Blue = cases documenting stop gained mutations, Green = missense SNPs in case 2, Purple = missense SNPs in case 9, Brown = missense SNPs of case 19.

**Figure 3 genes-13-01231-f003:**
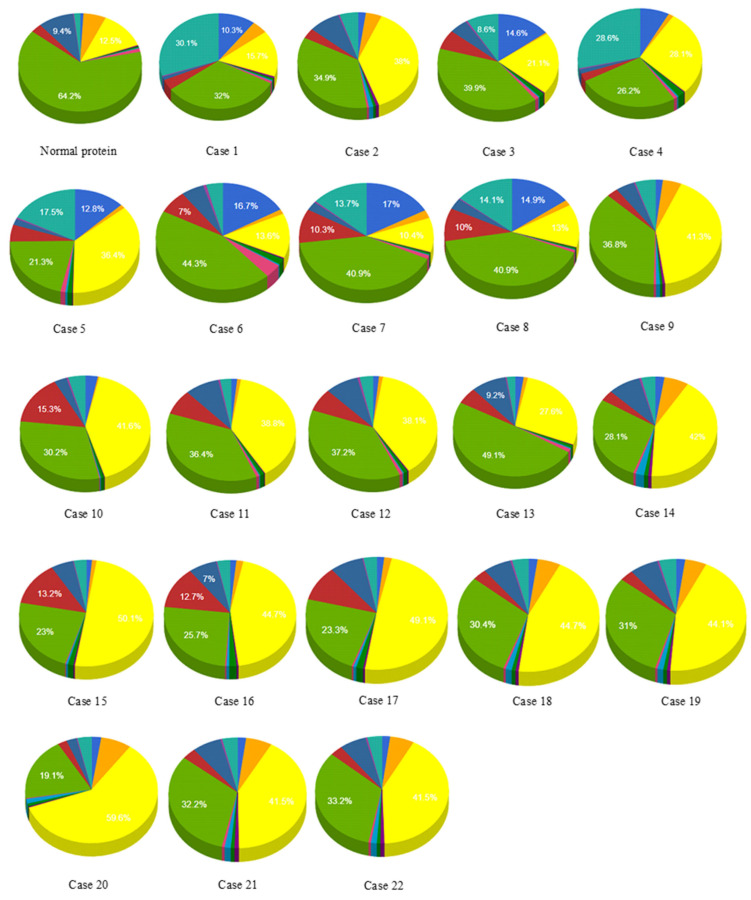
Effect of SNPs documented in cases 1 to 22 on the localization of mutant proteins. Purple = cytoskeletal, pink = lysosomal, dark green = endoplasmic reticulum, red = chloroplast.

**Figure 4 genes-13-01231-f004:**
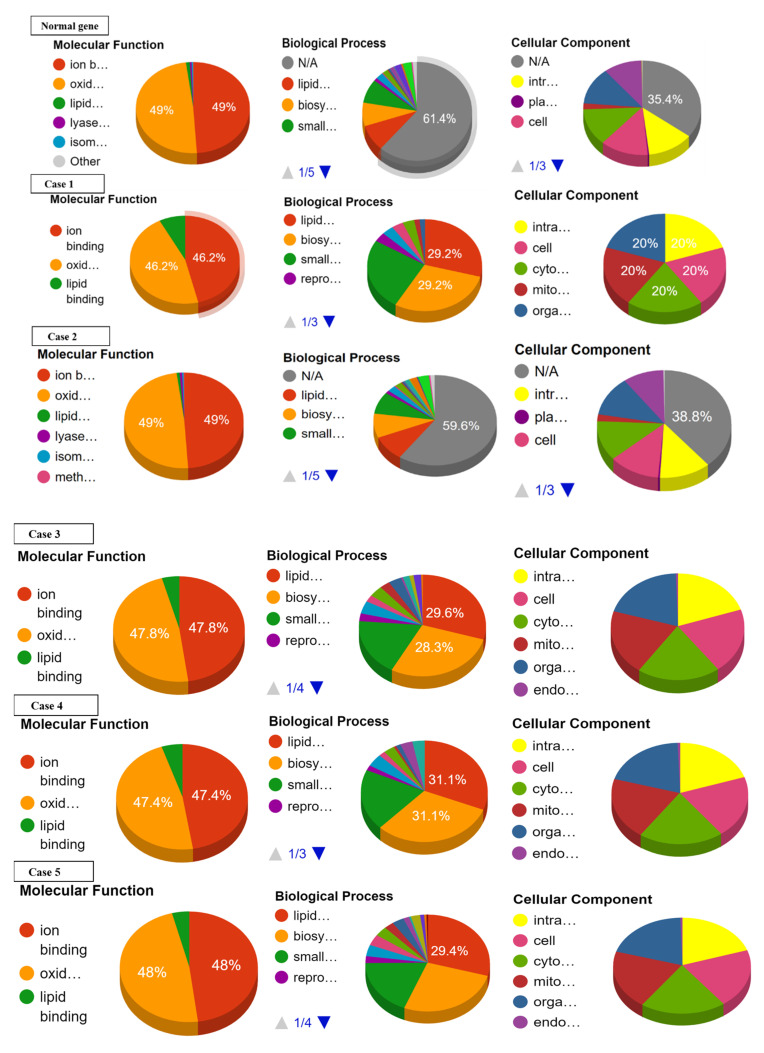
Effect of SNPs documented in present study in cases 1–16 on the ontology of mutated proteins.

**Figure 5 genes-13-01231-f005:**
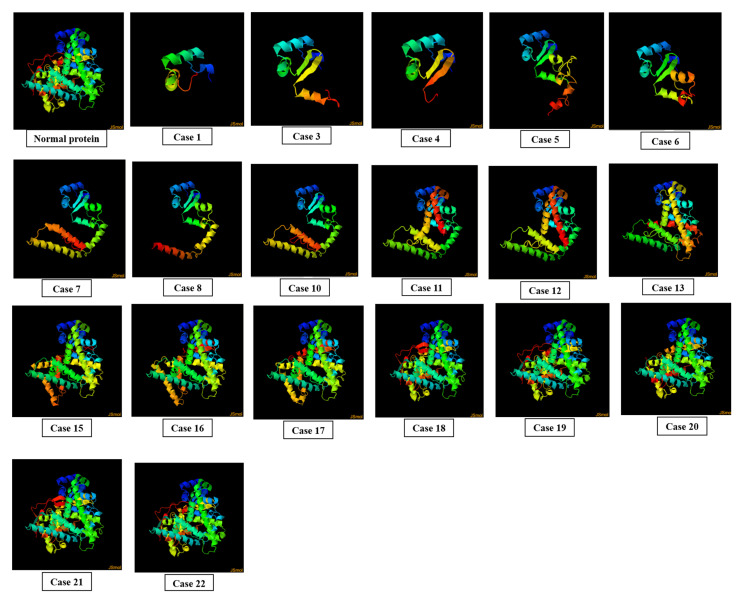
Effect of SNPs documented in cases 1, 3–8, 10–13, and 15–22 on the 3D structure of mutated proteins.

**Figure 6 genes-13-01231-f006:**
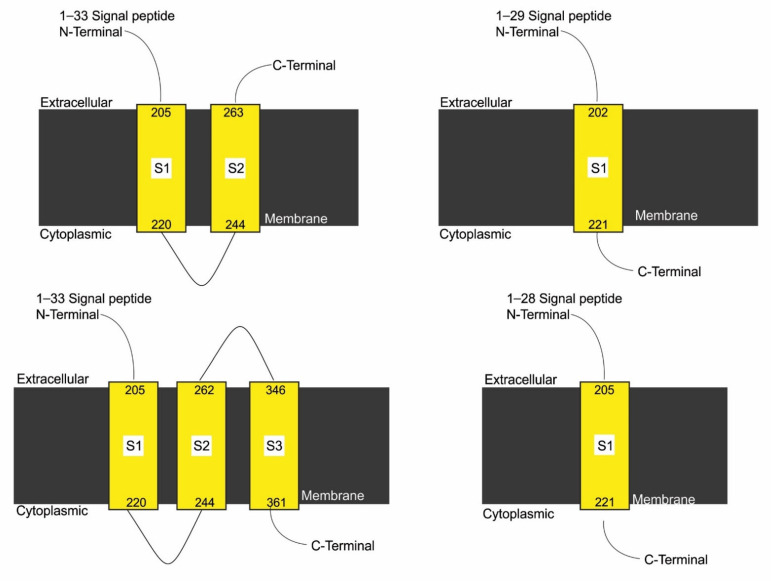
Effect of SNPs documented in cases 7, 9, and 10 on the topology of mutated protein.

**Table 1 genes-13-01231-t001:** SNPs of the CYP11A1 gene retrieved from the Ensemble database, their consequence type, and categorization of cases.

SNPs #	Region of Gene	SNP ID	Nucleotide Change	Case	Consequence Type	Amino Acid Change	Amino Acid Position	SIFT	Polyphen
1	Exon 1	rs750026801	tAt > tt	1	Frame shift mutation	Y > X	85	---	---
2		rs999606408	Ccc > Tcc	2	Missense mutation	P > S	7	0.45	0
3		rs1444841908	Ctg > Atg			L > M	20	0.01	0.621
4		rs533078157	aTc > aCc			I > T	40	0.39	0.003
5		rs1402720332	AaC > aaA			N > K	61	0.04	0.999
6		rs746842413	ttC > ttA			F > L	82	0	0.986
7		rs143655263	Gtc > Atc			V > I	79	0.44	0
8		rs1316467116	Cta > Gta			L > V	60	0.24	0.018
9		rs1471344818	tCc > tTc			S > F	41	0.04	0.838
10	Exon 2	rs546976108	Gag > Aag			E > K	91	0	1
11		rs190764523	Ggc > Agc			G > S	94	0	0.998
12		rs866217710	Gat > Aat			D > N	106	0.02	0.981
13	Exon 3	rs1258660118	Tcg > Acg			S > T	144	0.01	0.096
14		rs1393077247	aAg > aGg			K > R	148	1	0.075
15		rs1440123283	gAg > gGg			E > G	162	0.2	0.006
16		rs762299364	cTg > cCg			L > P	182	0	0.999
17		rs1400474918	Gag > Aag			E > K	208	0	0.952
18	Exon 2	rs776056840	Cga > Tga	3	Stop gained mutation	R > *	120	---	---
19		rs779154292	atC > at	4	Frame shift mutation	I > X	102	---	---
20		rs1217014229	Cac > CCac	5		H > PX	130	---	---
21	Exon 3	rs549043326	tCg > tAg	6	Stop gained mutation	S > *	144	---	---
22		rs1421587886	GcA > gc	7	Frame shift mutation	A > X	173	---	---
23		rs1178589612	cTg > cg	8		L > X	170	---	---
24	Exon 4	rs1567052760	gCc > gTc	9	Missense mutation	A > V	230	0.03	0.872
25		rs1207802955	Atc > Gtc			I > V	237	1	0.007
26		rs1461423064	cTc > cCc			L > P	248	0	0.998
27		rs1416104210	ttC > ttG			F > L	258	0.9	0.001
28		rs1596159786	Acc > Gcc			T > A	262	0.24	0.006
29		rs1228084259	Gct > Tct			A > S	268	0.62	0.506
30		rs1021942880	gCa > gTa			A > V	269	0	0.989
31	Exon 5	rs746124429	tTc > tCc			F > S	284	0.21	0.509
32		rs1459305798	Tgg > Cgg			W > R	286	1	0.001
33		rs1596159365	tAc > tCc			Y > S	298	0	0.995
34		rs747101738	Ctg > Gtg			L > V	324	0.01	0.286
35		rs1402190131	gCa > gAa			A > E	325	0	0.991
36		rs748120824	gAc > gGc			D > G	329	0	0.98
37		rs1424340465	aCg > aTg			T > M	330	0	0.997
38	Exon 6	rs139449608	aCg > aTg			T > M	331	0	1
39		rs1046646548	aTg > aCg			M > T	333	0.11	0.481
40		rs1567051424	tAt > tTt			Y > F	340	0.53	0.991
41		rs531531464	Gag > Aag			E > K	341	0	1
42		rs745719036	cAg > cGg			Q > R	372	0.05	0.051
43	Exon 4	rs755186597	Cga > Tga	10	Stop gained mutation	R > *	232	---	---
44	Exon 5	rs1224774813	Cag > Tag	11		Q > *	282	---	---
45		rs757299093	Ata > ta	12	Frame shift mutation	I > Y	279	---	---
46		rs1555425667	Gag > ag	13		E > X	314	---	---
47	Exon 7	rs867506250	Cac > Tac	14	Missense mutation	H > Y	388	0.43	0.938
48		rs1208632679	Tcc > Ccc			S > P	391	0.01	0.437
49		rs752776256	Cag > Aag			Q > K	395	0	0.967
50		rs1220602604	Ctt > Gtt			L > V	398	1	0.037
51		rs200029503	Gta > Tta			V > L	399	0.12	0.012
52		rs1398357296	Gtt > Ttt			V > F	403	0	0.869
53		rs1321165216	Ctt > Att			L > I	404	0.29	0.239
54		rs944327325	Gat > Cat			D > H	406	0	0.748
55		rs771663597	aTt > aCt			I > T	409	0	0.656
56		rs1448075161	Gcc > Acc			A > T	411	0.03	0.409
57	Exon 8	rs1344040376	Caa > Gaa			Q > E	416	0.04	0.986
58		rs1416463682	tAt > tGt			Y > C	420	0	0.998
59		rs780138488	Cca > Tca			P > S	437	0	1
60		rs774799229	cTg > cCg			L > P	463	0	0.992
61		rs757055824	Gct > Act			A > T	468	0	1
62		rs758061011	Atg > Gtg			M > V	472	0	0.977
63		rs1326008763	Atc > Gtc			I > V	474	0.01	0.135
64	Exon 9	rs566280511	Atg > Ctg			M > L	479	0.18	0.011
65		rs754698583	Gtt > Att			V > I	485	1	0.007
66		rs775664050	gTg > gAg			V > E	493	0	0.997
67		rs551306530	aTg > aCg			M > T	502	0.17	0.085
68		rs148124218	Ccc > Tcc			P > S	513	0	0.999
69		rs777925426	Ttt > Ctt			F > L	514	1	0
70		rs200726137	aAc > aTc			N > I	515	0.06	0.127
71	Exon 7	rs1454328072	Cga > Tga	15	Stop gained mutation	R > *	405	---	---
72	Exon 8	rs762412759	Cga > Tga	16		R > *	424	---	---
73		rs755975808	Cga > Tga	17		R > *	439	---	---
74	Exon 9	rs754610565	Cag > Tag	18		Q > *	521	---	---
75		rs779413653	Tga > Aga	19		R > *	522	---	---
76		rs765916701	Caa > Taa	20		Q > *	488	---	---
77		rs1368450780	Cag > Tag	21		Q > *	516	---	---
78		rs747901197	Gaa > Taa	22		E > *	517	---	---

Symbol * shown in cases of stop gained mutations is used to represent absence of amino acid.

**Table 2 genes-13-01231-t002:** Prediction of SNPs’ effect on the stability of protein through I Mutant Suite.

#	SNP ID	DDG (kcal/mole)	Stability	RI
1	rs999606408	−0.47	Decrease	5
2	rs1444841908	−1.33	Decrease	8
3	rs533078157	−2.24	Decrease	9
4	rs1402720332	−0.31	Decrease	5
5	rs746842413	−1.20	Decrease	8
6	rs143655263	−0.21	Decrease	4
7	rs1316467116	−1.62	Decrease	8
8	rs1471344818	0.03	Increase	3
9	rs546976108	−0.89	Decrease	8
10	rs190764523	−1.53	Decrease	9
11	rs866217710	−1.16	Decrease	5
12	rs1258660118	0.09	Increase	5
13	rs1393077247	0.07	Increase	3
14	rs1440123283	−1.04	Decrease	8
15	rs762299364	−1.80	Decrease	7
16	rs1400474918	−1.08	Decrease	8
17	rs1567052760	−0.35	Decrease	4
18	rs1207802955	−1.00	Decrease	9
19	rs1461423064	−1.65	Decrease	3
20	rs1416104210	−1.63	Decrease	8
21	rs1596159786	−1.31	Decrease	9
22	rs1228084259	−0.74	Decrease	9
23	rs1021942880	−0.19	Decrease	2
24	rs746124429	−1.93	Decrease	9
25	rs1459305798	−1.16	Decrease	9
26	rs1596159365	−1.41	Decrease	6
27	rs747101738	−1.47	Decrease	8
28	rs1402190131	−0.42	Decrease	3
29	rs748120824	−0.40	Decrease	2
30	rs1424340465	0.24	Increase	2
31	rs139449608	0.13	Increase	1
32	rs1046646548	−0.67	Decrease	5
33	rs1567051424	0.08	Increase	3
34	rs531531464	−0.69	Decrease	7
35	rs745719036	−0.05	Decrease	1
36	rs867506250	0.22	Increase	5
37	rs1208632679	−0.12	Increase	4
38	rs752776256	−0.56	Increase	1
39	rs1220602604	−1.75	Decrease	7
40	rs200029503	−1.63	Decrease	8
41	rs1398357296	−1.61	Decrease	8
42	rs1321165216	−1.52	Decrease	9
43	rs944327325	−0.96	Decrease	9
44	rs771663597	−2.38	Decrease	9
45	rs1448075161	−0.67	Decrease	7
46	rs1344040376	−0.22	Decrease	1
47	rs1416463682	−1.03	Increase	1
48	rs780138488	−1.58	Decrease	8
49	rs774799229	−1.62	Decrease	6
50	rs757055824	−0.69	Decrease	6
51	rs758061011	−0.69	Decrease	8
52	rs1326008763	−0.64	Decrease	6
53	rs566280511	−0.94	Decrease	7
54	rs754698583	−0.73	Decrease	8
55	rs775664050	−1.59	Decrease	7
56	rs551306530	−1.06	Decrease	6
57	rs148124218	−1.15	Decrease	6
58	rs777925426	−0.48	Decrease	0
59	rs200726137	1.23	Increase	7

DDG = measurement of protein stability in terms of free energy change, RI = reliability index.

**Table 3 genes-13-01231-t003:** Effect of SNPs in cases 1 to 22 on the localization of mutated proteins.

Case ID	M	C	N	P	E	PM	GA	ER	CT	L	V
Normal	3.208	0.624	0.085	0.469	0.047	0.297	0.013	0.025	0.016	0.057	0.011
Case 1	1.602	0.784	1.506	0.057	0.515	0.218	0.018	0.036	0.030	0.038	0.017
Case 2	1.747	1.899	0.250	0.424	0.108	0.214	0.060	0.036	0.041	0.028	0.021
Case 3	1.995	1.057	0.429	0.246	0.728	0.023	0.017	0.078	0.016	0.057	0.026
Case 4	1.311	1.407	1.429	0.072	0.386	0.069	0.020	0.099	0.012	0.048	0.019
Case 5	1.063	1.820	0.877	0.099	0.638	0.060	0.026	0.060	0.009	0.061	0.022
Case 6	2.213	0.678	0.203	0.277	0.834	0.072	0.018	0.095	0.017	0.206	0.038
Case 7	2.046	0.521	0.684	0.133	0.848	0.124	0.011	0.031	0.009	0.057	0.021
Case 8	2.047	0.650	0.707	0.168	0.746	0.085	0.009	0.028	0.007	0.039	0.015
Case 9	1.839	2.064	0.252	0.232	0.093	0.230	0.050	0.028	0.032	0.021	0.019
Case 10	1.510	2.082	0.216	0.154	0.159	0.019	0.011	0.042	0.007	0.010	0.024
Case 11	1.822	1.939	0.141	0.431	0.079	0.046	0.012	0.058	0.010	0.040	0.025
Case 12	1.859	1.907	0.170	0.419	0.079	0.050	0.014	0.055	0.014	0.044	0.032
Case 13	2.457	1.380	0.112	0.461	0.114	0.055	0.017	0.051	0.010	0.058	0.018
Case 14	1.404	2.009	0.184	0.426	0.124	0.320	0.109	0.053	0.044	0.034	0.028
Case 15	1.150	2.509	0.154	0.270	0.071	0.056	0.024	0.067	0.018	0.011	0.011
Case 16	1.283	2.236	0.162	0.349	0.080	0.086	0.034	0.082	0.023	0.017	0.012
Case 17	1.165	2.455	0.155	0.381	0.088	0.087	0.033	0.079	0.022	0.019	0.014
Case 18	1.522	2.235	0.189	0.321	0.108	0.264	0.069	0.044	0.036	0.027	0.022
Case 19	1.552	2.205	0.200	0.329	0.109	0.248	0.068	0.043	0.038	0.027	0.022
Case 20	0.954	2.980	0.176	0.124	0.122	0.374	0.073	0.047	0.015	0.009	0.013
Case 21	1.609	2.076	0.184	0.341	0.104	0.307	0.072	0.043	0.045	0.033	0.022
Case 22	1.659	2.077	0.182	0.323	0.103	0.293	0.071	0.043	0.042	0.030	0.022

M = mitochondrial; C = cytoplasmic; N = nucleus; P = peroxisomal; PM = plasma membrane; GA = golgi apparatus; ER = endoplasmic reticulum; CT = cytoskeletal; L = lysosomal; V = vacuolar.

**Table 4 genes-13-01231-t004:** Effect of SNPs on the ontology of mutated proteins.

Case ID	Molecular Functions			Biological Process		
	Ion Binding	Oxidoreductase Activity	Lipid Binding	Lipid Metabolism	Biosynthesis	Small Molecule Metabolism
Normal gene	48.9	48.9	0.9	9.1	8.8	8.1
1	46.2	46.2	7.7	29.2	29.2	25.3
2	49	49	0.8	8.9	8.7	7.9
3	47.8	47.8	4.3	29.6	28.3	18.3
4	47.4	47.4	5.1	31.1	31.1	19.5
5	48	48	4	29.4	26.6	18.8
6	48	48	4	9.8	26.6	18.8
7	48.1	48.1	3.7	28.8	25.3	19.1
8	48.2	48.2	3.6	28	24.6	18.6
9	49	49	0.6	9.8	9.1	8.6
10	48.6	48.6	2.7	23.1	19.6	16
11	48.7	48.7	2.7	22.7	19.2	15.7
12	48.7	48.7	2.7	22.7	19.2	15.7
13	48.7	48.7	2.7	22.7	19.2	15.7
14	49	49	0.9	9	8.3	7.8
15	49	49	2.1	18.2	15	12.8
16	49.1	49.1	1.7	14.8	10.4	11.2
17	49.5	49.5	0.9	9.6	7.6	8
18	48.9	48.9	0.9	9.1	8.8	8.1
19	48.9	48.9	0.9	9.1	8.8	8.1
20	48.9	48.9	0.9	9	8.7	8
21	48.9	48.9	0.9	9.1	8.8	8.1
22	48.9	48.9	0.9	9.1	8.8	8.1

**Table 5 genes-13-01231-t005:** Physicochemical characteristics of the normal and mutant CYP11A1 protein in different cases.

Case ID	No. of Amino Acids	Molecular Weight	Theoretical pI	Estimated Half Life	Instability Index	Aliphatic Index	Extinction Coefficient (M^−1^ cm^−1^)
Normal	553	63,785.85	5.83	30 h	31.19	83.91	87,445
Case 1	144	16,063.12	10.52	30 h	51.45	54.86	23,490
Case 2	555	64,033.25	6.35	30 h	31.57	82.20	87,445
Case 3	121	13,533.36	7.93	30 h	34.30	76.45	16,960
Case 4	202	22,095.05	9.33	30 h	52.01	70.89	22,585
Case 5	201	22,029.91	6.76	30 h	50.14	74.13	24,075
Case 6	143	16,271.72	9.59	30 h	32.86	83.08	26,930
Case 7	230	25,559.33	9.46	30 h	43.24	78.00	38,180
Case 8	232	25,978.79	9.56	30 h	43.58	73.53	43,680
Case 9	555	63,855.02	6.07	30 h	32.76	83.75	78,965
Case 10	237	26,787.50	7.86	30 h	30.55	81.81	33,920
Case 11	291	33,246.95	7.14	30 h	27.27	81.72	47,900
Case 12	297	33,847.53	7.85	30 h	26.10	78.75	47,900
Case 13	351	40,289.79	7.82	30 h	29.51	78.29	63,495
Case 14	547	63,085.11	6.06	30 h	33.03	85.36	87,445
Case 15	428	49,040.00	6.19	30 h	26.97	85.63	66,350
Case 16	449	51,374.76	6.19	30 h	28.30	87.06	69,330
Case 17	464	53,215.72	5.91	30 h	28.98	84.25	69,330
Case 18	552	63,657.72	5.83	30 h	30.88	84.06	87,445
Case 19	554	63,942.03	5.90	30 h	31.15	83.75	87,445
Case 20	517	59,607.16	6.02	30 h	31.13	85.22	81,945
Case 21	547	63,100.16	5.90	30 h	30.99	84.64	87,445
Case 22	548	63,288.29	5.90	30 h	30.82	84.49	87,445

## Data Availability

The SNPs documented in present study have been retrieved from ENSEMBL database ((https://web.expasy.org/translate).

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
