# Peer review of "In-Silico Investigation of Effects of Single-Nucleotide Polymorphisms in PCOS-Associated CYP11A1 Gene on Mutated Proteins"

_genes, 2022, doi:10.3390/genes13071231_

Round 1

Reviewer 1 Report

The manuscript presents work,  in analyzing the SNPs in CYP11A1 gene that can lead to PCOS. Effect of this mutations are analyzed using set of publicly available bioinformatics tools. The manuscript touches the effect of these mutations in many aspect like sub-cellular localization, ontology, instability index, 3-D and 2-D structures , transmembrane topology etc.  The current tools use for the analysis has been described and it could benefit research and analysis of other important genes associated with diseases. However, I would suggest the authors to address the following aspects:

(1) Give a schematic diagram of in- silico analysis scheme as a Figure 1 which will be easy to understand for the readers

2) Is there any relationship among the identified SNPs? Does it alter the protein- protein network interaction of CYP11A1 with other proteins?

(3) I suggest to extend the study to use multiple bioinformatics tools which is normally use in this type of SNPs study along with CELLO2GO,ProtParam and PHYRE2. Using multiple bioinformatics tool in studying different aspect of protein mutation will help the authors to filter or make the claim stronger (I suggest using tools like SIFT, PolyPhen2, PROVEAN,I mutant, ITASSER,  PROCHECK, MutPred2)

4) Elaborate more on the discussion or conclusion part. Current manuscript discussion or conclusion part is too short for all the analysis that has been done on effect of SNPs.

  5) Increase the resolution of current figures so that it will be easier to see.

Author Response

University of Punjab

School of Biochemistry & Biotechnology

 Quaid-e-Azam Campus Lahore, Pakistan

                                                                                                                                    June 27, 2022

Subject: Rebutal Letter (Reviewer 1)

Reviewers comment 1:

                 No a specific SNP to explain the high excess of androgen in PCOS.

Reply:

CYP11A1 gene encodes for enzyme cholesterol desmolase which catalyzes the transformation of cholesterol into pregnenolone (a precursor of steroid hormones) in mitochondria. This is the very first step of steroidogenesis pathway. Any disturbance in this pathway may result in abnormal level of steroid hormones in body.

Single nucleotide polymorphisms in a gene might affect in multiple ways. Like they may cause the up-regulation or down-regulation of the gene. They may also cause the formation of a protein with improper structure leading to abnormal functioning. This project is an attempt to explore the mutations that can lead to any of these changes ultimately causing hormonal disturbances. Variations with adverse effects on 3D structure, physicochemical attributes or sub-cellular localization of protein might be the contributing to PCOS. This study has provided us an idea about the possible SNPs that are definitely disturbing the normal steroid hormones synthesis pathway in female body and hence, provides a strong base for future experimentation. Now, in order to find out the association of an individual SNP with high level of androgens, experiments involving control and experimental groups can be designed. Therefore, this project has enabled us to find out majority of the possible biomarkers for PCOS prognosis and diagnosis.

Reviewers comment 2:

                 Similarly, the SNP so far could not to explain the insulin resistance in PCOS.

Reply:

            PCOS is a multifactorial trait which is controlled by multiple genes. The focus of present study is CYP11A gene that is definitely associated with PCOS as it regulates steroidogenic hormones synthesis. This study was not an attempt to determine the SNPs that might be related with all the symptoms of PCOS like insulin resistance, hirsutism and irregular or no ovulation etc. Even the excessive research on this disease has not yet solved the mystery of PCOS definite causes. And the exact cause of PCOS is still unknown. In this scenario, we have tried our best to sort out this mystery by providing the variations that are associated with steroid hormonal disturbance which might help in detecting the probability of developing PCOS in females in future. Another fact is that insulin resistance is itself under the control of multiple genes and environmental factors. In order to identify individual SNP associated with insulin resistance in PCOS patients, a long range of experiments involving the experimental group comprising of PCOS patients with high insulin resistance and the control groups with PCOS patients with low insulin resistance or healthy females with insulin resistance are required. Because as per the literature, in many cases insulin resistance in healthy females is also of the same level as in PCOS patients (Cibula, 2004). The comparative analysis through long term experimentation can be helpful, only to some extent, in identification of insulin resistance associated SNPs.

Reference:

Cibula, D. Is insulin resistance an essential component of PCOS? The influence of confounding factors. Human Reproduction 2004, 19, 757-759.

Dr. Fatima Muccee

Assistant Professor

School of Biochemistry & Biotechnology

Reviewer 2 Report

This is an interesting study to investigate SNP in PCOS. It could provided some informations regarding the mechanisms of PCOS. Some comments about this study:

1. No a specific SNP to explain the high excess of androgen in PCOS.

2. Similarly, the SNP so far could not to explain the insulin resistance in PCOS.

Author Response

University of Punjab

School of Biochemistry & Biotechnology

 Quaid-e-Azam Campus Lahore, Pakistan

                                                                                                                                                                              June 27 2022

Subject: Rebutal Letter (Reviewer 2

Reviewers comment 1:

Give a schematic diagram of in-silico analysis scheme as a Figure 1 which will be easy to understand for the readers.

Reply

            A schematic diagram of in-silico scheme as Figure 1 has been incorporated in manuscript in section 2 (Materials and Methods).

Figure 1: A schematic representation of in-silico analysis performed in present study

Reviewers comment 2

Is there any relationship among the identified SNPs? Does it alter the protein- protein network interaction of CYP11A1 with other proteins?

Reply

The relationship between SNPs documented in present study is not the focus of this project. Each SNP has been independently analyzed. However, several SNPs were found to have similarity in their effects on different attributes of encoded proteins.

As far as, the protein-protein interaction of CYP11A1 protein is concerned, not a single SNP was found to have any impact. The effect was confirmed through Search Tool for   the Retrieval of Interacting Genes i.e. STRING. As, the SNPs did not alter protein-protein interactions of CYP11A1 protein so the data has not been incorporated in manuscript. However, STRING results in some cases are attached as a proof (Figure A) in rebuttal letter for the kind information of reviewer.

Figure A: Analysis of interaction of CYP11A1 protein with different other proteins through STRING for different cases

Reviewers comment 3:

I suggest to extend the study to use multiple bioinformatics tools which is normally use in this type of SNPs study along with CELLO2GO, ProtParam and PHYRE2. Using multiple bioinformatics tool in studying different aspect of protein mutation will help the authors to filter or make the claim stronger (I suggest using tools like SIFT, PolyPhen2, PROVEAN,I mutant, ITASSER,  PROCHECK, MutPred2).

Reply:

            According to suggestion of Reviewer, we have added more bioinformatics tools i.e. SIFT and PolyPhen2 (SUBSECTION 2.2 and 3.8 and Table 1) and I mutant in (SUBSECTION 2.4 and 3.1 and Table 2) in order to determine the deleteriousness and effect of mutations on stability of mutated  proteins, respectively. Changes incorporated are highlighted yellow.

According to SIFT, mutations rs750026801, rs533078157, rs143655263, rs1316467116, rs1393077247, rs1440123283, rs1207802955, rs1416104210, rs1596159786, rs1228084259, rs746124429, rs1459305798, rs1046646548, rs1567051424, rs867506250, rs1220602604, rs200029503, rs1321165216, rs566280511, rs754698583, rs551306530, rs777925426 and rs200726137 are tolerated. While the SNPs rs1444841908, rs1402720332, rs746842413, rs1471344818, rs546976108, rs190764523, rs866217710, rs1258660118, rs762299364, rs1400474918, rs1567052760, rs1461423064, rs1021942880, rs1596159365, rs747101738, rs1402190131, rs748120824, rs1424340465, rs139449608, rs531531464, rs745719036, rs867506250, rs1208632679, rs752776256, rs1398357296, rs944327325, rs771663597, rs1448075161, rs1344040376, rs1416463682, rs780138488, rs774799229, rs757055824, rs758061011, rs1326008763, rs775664050, rs750026801, rs776056840, rs779154292, rs1217014229, rs549043326, rs1421587886, rs1178589612, rs755186597, rs1224774813, rs757299093, rs1555425667, rs762412759 , rs755975808, rs754610565, rs779413653, rs765916701, rs1368450780, rs747901197  and rs148124218 are found to have deleterious effect.

According to PolyPhen, mutations rs750026801, rs533078157, rs143655263, rs1316467116, rs1258660118, rs1393077247, rs1440123283, rs1207802955, rs1416104210, rs1596159786, rs1459305798, rs747101738, rs745719036, rs1208632679, rs1220602604, rs200029503, rs1321165216, rs1448075161, rs566280511, rs754698583, rs551306530, rs777925426 and rs200726137 are categorized as benign. While the SNPs rs1444841908, rs1402720332, rs746842413, rs1471344818, rs546976108, rs190764523, rs866217710, rs762299364, rs1400474918, rs1567052760, rs1461423064, rs1228084259, rs1021942880, rs746124429, rs1596159365, rs1402190131, rs748120824, rs1424340465, rs139449608, rs1046646548, rs1567051424, rs531531464, rs752776256, rs1398357296, rs944327325, rs771663597, rs1344040376, rs1416463682, rs780138488, rs774799229, rs757055824, rs758061011, rs1326008763, rs775664050, rs750026801, rs776056840, rs779154292, rs1217014229, rs549043326, rs1421587886, rs1178589612, rs755186597, rs1224774813, rs757299093, rs1555425667, rs762412759 , rs755975808, rs754610565, rs779413653, rs765916701, rs1368450780, rs747901197 and rs148124218 are possibly damaging.

Table 1: SNPs of CYP11A1 gene retrieved from Ensemble database, their consequence type and categorization of cases

SNPs #

Region of gene

SNP ID

Nucleotide change

Case

Consequence type

Amino acid change

Amino acid position

SIFT

Poly-phen

1

Exon 1

rs750026801

tAt>tt     

1

Frame shift mutation

Y>X

85

---

---

2

rs999606408

Ccc>Tcc   

2

Missense mutation

P>S

7

0.45

0

3

rs1444841908 

Ctg>Atg

L>M

20

0.01

0.621

4

rs533078157

aTc>aCc

I>T

40

0.39

0.003

5

rs1402720332 

AaC>aaA

N>K

61

0.04

0.999

6

rs746842413

ttC>ttA

F>L

82

0

0.986

7

rs143655263

Gtc>Atc

V>I

79

0.44

0

8

rs1316467116 

Cta>Gta

L>V

60

0.24

0.018

9

rs1471344818

tCc>tTc

S>F

41

0.04

0.838

10

 Exon 2

rs546976108

Gag>Aag

E>K

91

0

1

11

rs190764523

Ggc>Agc

G>S

94

0

0.998

12

rs866217710

Gat>Aat

D>N

106

0.02

0.981

13

Exon 3

rs1258660118

Tcg>Acg

S>T

144

0.01

0.096

14

rs1393077247

aAg>aGg

K>R

148

1

0.075

15

rs1440123283

gAg>gGg

E>G

162

0.2

0.006

16

rs762299364

cTg>cCg

L>P

182

0

0.999

17

rs1400474918

Gag>Aag

E>K

208

0

0.952

18

Exon 2

rs776056840

Cga>Tga  

3

Stop gained mutation

R>*

120

---

---

19

rs779154292

atC>at  

4

Frame shift mutation

I>X

102

---

---

20

rs1217014229

Cac>CCac  

5

H>PX

130

---

---

21

Exon 3

rs549043326

tCg>tAg  

6

Stop gained mutation

S>*

144

---

---

22

rs1421587886

GcA>gc  

7

Frame shift mutation

A>X

173

---

---

23

rs1178589612

cTg>cg  

8

L>X

170

---

---

24

Exon 4

rs1567052760

gCc>gTc  

  9

Missense mutation

A>V

230

0.03

0.872

25

rs1207802955 

Atc>Gtc

I>V

237

1

0.007

26

rs1461423064

cTc>cCc

L>P

248

0

0.998

27

rs1416104210 

ttC - ttG

F>L

258

0.9

0.001

28

rs1596159786

Acc>Gcc

T>A

262

0.24

0.006

29

rs1228084259

Gct>Tct

A>S

268

0.62

0.506

30

31

rs1021942880

gCa>gTa

A>V

269

0

0.989

Exon 5

rs746124429

tTc>tCc

F>S

284

0.21

0.509

32

rs1459305798 

Tgg>Cgg

W>R

286

1

0.001

33

rs1596159365 

tAc>tCc

Y>S

298

0

0.995

34

rs747101738

Ctg>Gtg

L>V

324

0.01

0.286

35

rs1402190131

gCa>gAa

A>E

325

0

0.991

  36

rs748120824

gAc>gGc

D>G

329

0

0.98

37

rs1424340465 

aCg>aTg

T>M

330

0

0.997

38

Exon 6

rs139449608 

aCg>aTg

T>M

331

0

1

39

rs1046646548

aTg>aCg

M>T

333

0.11

0.481

40

rs1567051424

tAt>tTt

Y>F

340

0.53

0.991

41

rs531531464 

Gag>Aag

E>K

341

0

1

42

rs745719036

cAg>cGg

Q>R

372

0.05

0.051

43

Exon 4

rs755186597

Cga>Tga   

10

Stop gained mutation

R>*

232

---

---

44

Exon 5

rs1224774813

Cag>Tag  

11

Q>*

282

---

---

45

rs757299093

Ata>ta  

12

Frame shift mutation

I> -

279

---

---

46

rs1555425667

Gag>ag  

13

E>X

314

---

---

47

Exon 7

rs867506250 

Cac>Tac  

14

Missense mutation

H>Y

388

0.43

0.938

48

rs1208632679 

Tcc>Ccc

S>P

391

0.01

0.437

49

rs752776256

Cag>Aag

Q>K

395

0

0.967

50

rs1220602604

Ctt>Gtt

L>V

398

1

0.037

51

rs200029503

Gta>Tta

V>L

399

0.12

0.012

52

rs1398357296

Gtt>Ttt

V>F

403

0

0.869

53

rs1321165216 

Ctt>Att

L>I

404

0.29

0.239

54

rs944327325

Gat>Cat

D>H

406

0

0.748

55

rs771663597

aTt>aCt

I>T

409

0

0.656

56

rs1448075161

Gcc>Acc

A>T

411

0.03

0.409

57

Exon8

rs1344040376

Caa>Gaa

Q>E

416

0.04

0.986

58

rs1416463682

tAt>tGt

Y>C

420

0

0.998

59

rs780138488

Cca>Tca

P>S

437

0

1

60

rs774799229

cTg>cCg

L>P

463

0

0.992

61

rs757055824

Gct>Act

A>T

468

0

1

62

rs758061011

Atg>Gtg

M>V

472

0

0.977

63

rs1326008763

Atc>Gtc

I>V

474

0.01

0.135

64

Exon 9

rs566280511

Atg>Ctg

M>L

479

0.18

0.011

65

rs754698583

Gtt>Att

V>I

485

1

0.007

66

rs775664050

gTg>gAg

V>E

493

0

0.997

67

rs551306530

aTg>aCg

M>T

502

0.17

0.085

68

rs148124218

Ccc>Tcc

P>S

513

0

0.999

69

rs777925426 

Ttt>Ctt

F>L

514

1

0

70

rs200726137

aAc>aTc

N>I

515

0.06

0.127

71

Exon 7

rs1454328072

Cga>Tga  

15

Stop gained mutation

R>*

405

---

---

72

Exon 8

rs762412759 

Cga>Tga  

16

R>*

424

---

---

73

rs755975808

Cga>Tga 

17

R>*

439

---

---

74

Exon 9

rs754610565 

Cag>Tag 

18

Q> *

521

---

---

75

rs779413653

Tga>Aga 

19

*>R

522

---

---

76

rs765916701

Caa>Taa  

20

Q>*

488

---

---

77

rs1368450780

Cag>Tag  

21

Q>*

516

---

---

78

rs747901197

Gaa>Taa  

22

E>*

517

---

---

The results of I Mutant are as follows;

3.1. Effect of SNPs on mutated protein stability

      Fifty-nine missense SNPs addressed in present study were examined for their influence on stability of proteins. It was found that all the missense mutations except rs1258660118, rs1393077247, rs1424340465, rs139449608, rs867506250, rs1208632679, rs752776256, rs1416463682 and rs200726137 were contributing to destabilization of corresponding mutated proteins (Table 2).

      Table 2. Prediction of SNPs effect on stability of protein through I Mutant Suite

#

SNP ID

DDG (kcal/mole)

Stability

RI

1

rs999606408

-0.47

Decrease

5

2

rs1444841908

-1.33

Decrease

8

3

rs533078157

-2.24

Decrease

9

4

rs1402720332

-0.31

Decrease

5

5

rs746842413

-1.20

Decrease

8

6

rs143655263

-0.21

Decrease

4

7

rs1316467116

-1.62

Decrease

8

8

rs1471344818

0.03

Increase

3

9

rs546976108

-0.89

Decrease

8

10

rs190764523

-1.53

Decrease

9

11

rs866217710

-1.16

Decrease

5

12

rs1258660118

0.09

Increase

5

13

rs1393077247

0.07

Increase

3

14

rs1440123283

-1.04

Decrease

8

15

rs762299364

-1.80

Decrease

7

16

rs1400474918

-1.08

Decrease

8

17

rs1567052760

-0.35

Decrease

4

18

rs1207802955

-1.00

Decrease

9

19

rs1461423064

-1.65

Decrease

3

20

rs1416104210

-1.63

Decrease

8

21

rs1596159786

-1.31

Decrease

9

22

rs1228084259

-0.74

Decrease

9

23

rs1021942880

-0.19

Decrease

2

24

rs746124429

-1.93

Decrease

9

25

rs1459305798

-1.16

Decrease

9

26

rs1596159365

-1.41

Decrease

6

27

rs747101738

-1.47

Decrease

8

28

rs1402190131

-0.42

Decrease

3

29

rs748120824

-0.40

Decrease

2

30

rs1424340465 

0.24

Increase

2

31

rs139449608 

0.13

Increase

1

32

rs1046646548

-0.67

Decrease

5

33

rs1567051424

0.08

Increase

3

34

rs531531464 

-0.69

Decrease

7

35

rs745719036

-0.05

Decrease

1

36

rs867506250 

0.22

Increase

5

37

rs1208632679 

-0.12

Increase

4

38

rs752776256

-0.56

Increase

1

39

rs1220602604

-1.75

Decrease

7

40

rs200029503

-1.63

Decrease

8

41

rs1398357296

-1.61

Decrease

8

42

rs1321165216 

-1.52

Decrease

9

43

rs944327325

-0.96

Decrease

9

44

rs771663597

-2.38

Decrease

9

45

rs1448075161

-0.67

Decrease

7

46

rs1344040376

-0.22

Decrease

1

47

rs1416463682

-1.03

Increase

1

48

rs780138488

-1.58

Decrease

8

49

rs774799229

-1.62

Decrease

6

50

rs757055824

-0.69

Decrease

6

51

rs758061011

-0.69

Decrease

8

52

rs1326008763

-0.64

Decrease

6

53

rs566280511

-0.94

Decrease

7

54

rs754698583

-0.73

Decrease

8

55

rs775664050

-1.59

Decrease

7

56

rs551306530

-1.06

Decrease

6

57

rs148124218

-1.15

Decrease

6

58

rs777925426

-0.48

Decrease

0

59

rs200726137

1.23

Increase

7

Reviewers comment 4:

Elaborate more on the discussion or conclusion part. Current manuscript discussion or conclusion part is too short for all the analysis that has been done on effect of SNPs.

Reply:

            The discussion and conclusion parts have been elaborated. Data from six different papers has been added and cited in Discussion part. Changes and references incorporated are highlighted yellow.

Reviewers comment 4:

Increase the resolution of current figures so that it will be easier to see.

Reply:

The resolution of figures has been increased to 600 dpi and incorporated in manuscript.

Dr. Fatima Muccee

Assistant Professor

School of Biochemistry & Biotechnology

Round 2

Reviewer 1 Report

Abstract can be modified with new analysis result and tools used

Figure 6 should be under the heading "Effect of SNPs on trans-membrane topology of mutated proteins "

Author Response

University of Punjab

School of Biochemistry & Biotechnology

 Quaid-e-Azam Campus Lahore, Pakistan

                                                                                                                                    July 1, 2022

Subject: Rebutal Letter (Reviewer 1)

Reviewers comment 1:

                 Abstract can be modified with new analysis result and tools used.

Reply:

The new analysis tools i.e. I-Mutant, SIFT and PolyPhen and the results of corresponding tools have been incorporated in Abstract through Track changes.

Abstract: PCOS is a reproductive disorder with multiple etiologies, mainly characterized by excess production of androgens. It is equally contributed by genes and environment. CYP11A1 gene is imperative for steroidogenesis so any dysregulation or mutation in this gene can lead to PCOS pathogenesis. Therefore, nucleotide diversity in this gene can be helpful in spotting likelihood of developing PCOS. Present study was initiated to investigate the effect of single nucleotide polymorphisms in human CYP11A1 gene on different attributes of encoded mutated proteins. i. e. sub-cellular localization, ontology, half-life, isoelectric point, instability index, aliphatic index, extinction coefficient, 3-D and 2-D structures and transmembrane topology.­­­ For this purpose, initially CDS and SNPs were retrieved for desired gene from Ensembl followed by translation of CDS using EXPASY tool. The protein sequence obtained was subjected to different tools including CELLO2GO, ProtParam, PHYRE2, I-Mutant, SIFT and PolyPhen. It was found that out of seventy-eight SNPs analyzed in this project, seventeen mutations i.e. rs750026801 in exon 1, rs776056840, rs779154292 and rs1217014229 in exon 2, rs549043326 in exon 3, rs755186597 in exon 4, rs1224774813, rs757299093 and rs1555425667 in exon 5, rs1454328072 in exon 7, rs762412759 and rs755975808 in exon 8 and rs754610565, rs779413653, rs765916701, rs1368450780 and rs747901197 in exon 9 considerably altered the structure, sub-cellular localization and physicochemical characteristics of mutated proteins. Among the fifty-nine missense SNPs documented in present study, fifty-five and fifty-three were found to be deleterious according to SIFT and PolyPhen tools, respectively. Forty-nine missense mutations were analyzed to have decreasing effect on stability of mutant proteins. Hence, these genetic variants can serve as potential biomarkers in human females for determining the probability of being predisposed to PCOS.

Reviewers comment 2:

Figure 6 should be under the heading "Effect of SNPs on trans-membrane topology of mutated proteins"

Reply:

            Figure 6 has been incorporated under the heading “Effect of SNPs on trans-membrane topology of mutated proteins"

Dr. Fatima Muccee

Assistant Professor

School of Biochemistry & Biotechnology
